# Predicting Response to Immunotargeted Therapy in Endometrial Cancer via Tumor Immune Microenvironment: A Multicenter, Observational Study

**DOI:** 10.3390/ijms25073933

**Published:** 2024-04-01

**Authors:** Anastasia Maltseva, Anna Kalinchuk, Nataliya Chernorubashkina, Virab Sisakyan, Igor Lots, Alina Gofman, Yulia Anzhiganova, Elizaveta Martynova, Ruslan Zukov, Elena Aleksandrova, Larisa Kolomiets, Liubov Tashireva

**Affiliations:** 1Cancer Research Institute, Tomsk National Research Medical Center, Russian Academy of Sciences, Tomsk 634050, Russia; maltseva.anast@gmail.com (A.M.); annakalinchuk2022@gmail.com (A.K.); kolomietsla@oncology.tomsk.ru (L.K.); 2Irkutsk Regional Oncology Center, 32 Frunze St., Irkutsk 664035, Russia; 3Novosibirsk Regional Clinical Oncology Center, 2 Plakhotnogo St., Novosibirsk 630108, Russia; sisakyan@mail.ru (V.S.); ig.y.lots@gmail.com (I.L.); 4Altai Regional Oncological Dispensary, 110 Zmeinogorsky tr., Barnaul 656000, Russia; alina-barnaul@mail.ru; 5Krasnoyarsk Regional Clinical Oncological Dispensary Named after A. I. Kryzhanovsky, 16 1-ya Smolenskaya St., Krasnoyarsk 660133, Russia; anzhi.yuliya@yandex.ru (Y.A.); zukov.ra@krasgmu.ru (R.Z.); 6Yakut Republican Oncology Center, Build. 1, 81 Stadukhina St., Yakutsk 677005, Russia

**Keywords:** endometrial cancer, immunotargeted therapy, tumor microenvironment, lymphocytes

## Abstract

Only one-third of patients with advanced MSS/pMMR endometrial cancer exhibit a lasting response to the combination treatment of Pembrolizumab and Lenvatinib. The combined administration of these two drugs is based on Lenvatinib’s ability to modulate the tumor microenvironment, enabling Pembrolizumab to exert its effect. These findings underscore the importance of exploring tumor microenvironment parameters to identify markers that can accurately select candidates for this type of therapy. An open non-randomized observational association study was conducted at six clinical centers, involving a total of 28 patients with advanced MSS/pMMR endometrial cancer who received Pembrolizumab and Lenvatinib therapy. Using TSA-associated multiplex immunofluorescence, we analyzed the proportion of CD8+ T lymphocytes, CD20+ B lymphocytes, FoxP3+ T regulatory lymphocytes, and CD163+ macrophages in tumor samples prior to immunotargeted therapy. The percentage of CD20+ B lymphocytes and the CD8-to-CD20 lymphocytes ratio was significantly higher in patients who responded to treatment compared to non-responders (responders vs. non-responders: 0.24 (0.1–1.24)% vs. 0.08 (0.00–0.15)%, *p* = 0.0114; 1.44 (0.58–2.70) arb. unit vs. 19.00 (3.80–34.78) arb. unit, *p* = 0.0031). The sensitivity and specificity of these biomarkers were 85.71% and 70.59%, and 85.71% and 85.71%, respectively. The proportion of CD20+ B lymphocytes and the CD8-to-CD20 lymphocytes ratio in the stroma of endometrial cancer serves as both a prognostic marker of response to immunotargeted therapy and a prognostic factor for progression-free survival in patients.

## 1. Introduction

According to Global Cancer Statistics, endometrial cancer ranked sixth in terms of incidence among all malignant tumors in women worldwide in 2020, with over 417 thousand new cases detected. From 2014 to 2018, there was a consistently high incidence of endometrial cancer, and the mortality rate increased by 1.9% from 2015 to 2019 [1]. The choice of treatment options for endometrial cancer progression depends on factors such as initial treatment, the extent of the malignant process, histological tumor type, and the patient’s condition [2]. In 2017, the FDA approved Pembrolizumab as a second-line therapy for MSI tumors, including endometrial cancer [3]. However, only about 30% of primary endometrial tumors exhibit microsatellite instability (MSI), while the remaining tumors (70%) are considered microsatellite stable (MSS) or have no defects in the repair system (pMMR) [4]. For these patients with MSS or pMMR status, the recommended treatment option is immunotargeted therapy, specifically a combination of Pembrolizumab and Lenvatinib, which received FDA accelerated approval in 2019 for the treatment of advanced endometrial tumors that are not MSI or dMMR [5]. However, the published results of the phase 3 LEAP-001 trial (NCT03884101) revealed that the trial did not meet its primary endpoints in the first line in patients with advanced or recurrent endometrial carcinoma whose disease is mismatch repair proficient (pMMR/MSS) [6]. This highlights significant challenges in patient stratification for this therapy and underscores the need for well-defined criteria in selecting patients for Pembrolizumab and Lenvatinib therapy.

However, only a third of patients show a durable response to the combination of Pembrolizumab and Lenvatinib, highlighting the need to identify criteria that can predict a long response in this patient group. Since the effects of this drug combination primarily involve immune mechanisms, various parameters of the tumor microenvironment could serve as potential predictive criteria. It should be noted that positive PD-L1 expression is not an indication for prescribing Pembrolizumab in endometrial cancer and does not have prognostic significance [7]. Lenvatinib’s primary antitumor mechanism of action is its antiangiogenic activity, achieved through the inhibition of vascular endothelial growth factor receptors (VEGFR1-3), fibroblast growth factor receptors (FGFR1-4), platelet-derived growth factor receptor α (PDGFR α), RET, and KIT [8]. Additionally, there is evidence of the immunomodulatory properties of angiogenesis inhibitors. In a mouse model, Lenvatinib significantly reduced the population of tumor-associated macrophages and increased the proportion of CD8-positive T cells, leading to the enhanced antitumor activity of PD-1 inhibitors [9,10]. In patients with hepatocellular carcinoma, recurrent tumors treated with Lenvatinib showed lower expressions of programmed death ligand 1 (PD-L1) and fewer Treg infiltrations compared to matched primary tumors [11]. So, Lenvatinib, used in combination, can potentially enhance Pembrolizumab effects and provide a more comprehensive approach to cancer treatment.

Therefore, understanding the functional heterogeneity of the tumor microenvironment is crucial for modulating the drug response to immunotherapy and its combinations, including the development of resistance. In the case of endometrial cancer, studying the features of the MSS microenvironment can help identify parameters that predict long-term responses to immunotargeted therapy.

## 2. Results

### 2.1. Patient Characteristics

All patients with recurrent or metastatic endometrial cancer (R/M EC) had MSS/pMMR status and received treatment with Pembrolizumab plus Lenvatinib. The treatment regimen consisted of Pembrolizumab administered intravenously at a dose of 200 mg once every 3 weeks, and Lenvatinib taken orally at a dose of 20 mg every day. The clinical and pathological parameters of the patients are presented in Table 1. The duration of clinical benefit (DoCB) was used to classify patients as responders (Rs) or non-responders (NRs) to Pembrolizumab plus Lenvatinib treatment [12]. Among the 28 patients, 19 were classified as responders, while nine were classified as non-responders. The DoCB is defined as the time from start treatment to disease progression or death in patients who achieve complete response, partial response, or stable disease for 24 weeks. It is a primary endpoint that is used in clinical trials in which disease stabilization in order to prolong survival is the primary goal. A total of nine patients experienced disease progression within 6 months of starting immunotargeted therapy. On the other hand, 15 patients showed a stable disease response, while four patients achieved a partial response. The DoCB in Rs were 14 months and in NRs, 6 months. The median progression-free survival (PFS) and overall survival (OS) were 12 and 47.5 months, respectively.

### 2.2. Immune Cells Composition of Tumor Microenvironment of Recurrent or Metastatic Endometrial Cancer

The composition of the tumor microenvironment in patients with recurrent or metastatic endometrial cancer was examined (Figure 1). Among the cell populations we analyzed, CD163+ macrophages (6.06 (4.24–9.14)%) were the most prevalent. Following in descending order were CD8+ T lymphocytes (1.59 (0.46–3.28)%), FoxP3+ T lymphocytes (1.19 (0.66–2.96)%), and CD20+ B lymphocytes (0.16 (0.06–1.10)%) (Friedman test with Dunn’s correction, *p* = 0.0001). While we did not specifically identify other cells in the tumor microenvironment, their morphology enabled us to differentiate fibroblasts, macrophages, and lymphocytes among them.

### 2.3. Levels of CD20+ B Lymphocytes and the CD8-to-CD20 Lymphocyte Ratio Are Associated with Response to Pembrolizumab plus Lenvatinib Treatment

To investigate the role of immune cells in patients with R/M EC undergoing treatment with Pembrolizumab plus Lenvatinib, we examined the infiltration of CD8+ T lymphocytes, CD20+ B lymphocytes, FoxP3+ T lymphocytes, and CD163+ macrophages within the tumor tissue of 28 cancer patients before immunotargeted therapy. The findings revealed a significant correlation between low levels of CD20+ B lymphocytes and the response of R/M EC patients to Pembrolizumab plus Lenvatinib treatment (*p* = 0.0220) (Figure 2A). Notably, the proportion of CD20+ B lymphocytes was significantly higher in patients who responded to treatment compared to non-responders (Rs vs. NRs; 0.24 (0.1–1.24)% vs. 0.08 (0.00–0.15)%, *p* = 0.0114). However, there was no significant association observed between CD8+ T lymphocytes, FoxP3+ T lymphocytes, CD163+ macrophages, and clinical response (CD8+: Rs vs. NRs: 1.33 (0.34–3.74)% vs. 1.63 (0.79–2.80)%, *p* = 0.7355; FoxP3+: Rs vs. NRs: 1.41 (0.72–3.58)% vs. 0.93 (0.23–2.10)%, *p* = 0.2587; CD163+: Rs vs. NRs: 6.40 (4.10–11.25)% vs. 5.85 (4.09–7.17)%, *p* = 0.3299) (Figure 2A).

We conducted ROC analysis to assess the predictive value of the CD20+ B lymphocyte proportion in determining the response to Pembrolizumab plus Lenvatinib treatment (Figure 2B). Using the ROC analysis, a CD20+ B lymphocyte proportion below 0.135% was found to be associated with a poor response to therapy in patients with R/M EC (AUC (95% CI) = 0.79 (0.61–0.97), *p* = 0.0242, sensitivity 85.71%, specificity 70.59%).

Furthermore, when examining the relationship of PD1-negative and PD1-positive CD20+ B lymphocyte levels with the response of R/M EC patients to Pembrolizumab plus Lenvatinib treatment, a significant difference in the percentage of CD20+ B lymphocytes was only observed within the PD1-negative subpopulation (*p* = 0.0220) (Figure 3A). Specifically, among patients who responded to treatment, the proportion of CD20+PD1− B lymphocytes was significantly higher compared to non-responders (Rs vs. NRs; 0.21 (0.07–1.67)% vs. 0.08 (0.00–0.15)%, *p* = 0.0401). However, there was no significant difference in the proportion of CD20+PD1+ B lymphocytes between responders and non-responders (Rs vs. NRs; 0.00 (0.00–0.14)% vs. 0.00 (0.00–0.00)%, *p* = 0.1282).

In order to evaluate the predictive significance of the proportion of CD20+PD1− B lymphocytes in determining the response to Pembrolizumab plus Lenvatinib treatment (Figure 3B), we conducted an ROC analysis. Our findings revealed that a CD20+PD1− B lymphocyte proportion below 0.135% was indicative of a negative response to the therapy, suggesting poor treatment outcomes. The area under the curve (AUC) was calculated to be 0.74 (95%CI: 0.55–0.92, *p* = 0.0412). The sensitivity of this marker was 77.78%, and the specificity was 63.16%.

Since the cells we studied belong to different types of immuno-inflammatory response, the CD8-to-CD20 lymphocytes ratio and CD8-to-FoxP3 lymphocytes ratio were calculated (Figure 4). As CD20+ B lymphocytes and FoxP3+ T lymphocytes were occasionally missing, preventing the calculation of the ratio, the number of observations in the groups was decreased. A significant difference in the CD8-to-CD20 lymphocytes ratio between responders and non-responders was found (Rs vs. NRs; 1.44 (0.58–2.70) arb. unit vs. 19.00 (3.80–34.78) arb. unit, *p* = 0.0031). Cut off was >3.366 to indicate a negative response to the therapy. The area under the curve (AUC) was calculated to be 0.74 (95%CI: 0.55–0.92, *p* = 0.0412). The sensitivity of this marker was 85.71%, and the specificity was 85.71%.

Considering the optimal balance of sensitivity and specificity, as well as the practicality of measuring each parameter in routine clinical practice, we chose the combined proportion of CD20+ B lymphocytes and the CD8-to-CD20 lymphocyte ratio as a predictive marker for further analysis.

### 2.4. CD20+ B Lymphocytes and the CD8-to-CD20 Lymphocytes Ratio as Prognostic Factors of PFS, but Not OS, in (R/M) EC Patients

The proportion of CD20+ B lymphocytes, the CD8-to-CD20 lymphocytes ratio, and clinical parameters such as FIGO, histological type, grade, myometrial invasion, lymphovascular invasion, carcinomatosis, and visceral metastases were further investigated using univariate non-linear simple regression analysis to prognosis progression during immunotargeted therapy (Table 2).

The risk of short DoCB as estimated by the odds ratio was highest for patients with CD20+ B lymphocyte values ≤ 0.135% (OR = 2.56; 95%CI: 1.12–6.80, *p* = 0.004) and patients with a CD8-to-CD20 lymphocytes ratio ≥ 3.366 (OR = 8.21; 95%CI: 2.76–23.10, *p* = 0.0001). Bias values evaluated by Bootstrap validation were close to zero for the proportion of CD20+ B lymphocytes and the CD8-to-CD20 lymphocytes ratio.

Also, all clinical and pathological parameters were further investigated using Cox regression univariate analysis to predict progression-free survival (Table 2). Patients with a CD20+ B lymphocytes percentage >0.135% and a CD8-to-CD20 lymphocytes ratio < 3.366 showed an increase in PFS (CD20+ > 0.135% vs. CD20+ ≤ 0.135%: median not reached vs. 14 months, *p* = 0.0336; CD8-to-CD20 lymphocytes ratio < 3.366 vs. CD8-to-CD20 lymphocytes ratio ≥ 3.366: median not reached vs. 6 months, *p* < 0.0001), but not OS (CD20+ > 0.135% vs. CD20+ ≤ 0.135%: median not reached vs. 80 months, *p* = 0.4328; CD8-to-CD20 lymphocytes ratio < 3.366 vs. CD8-to-CD20 lymphocytes ratio ≥ 3.366: median not reached vs. 80 months, *p* = 0.0867) (Figure 5).

In addition, grade, the proportion of CD20+ B lymphocytes, and the CD8-to-CD20 lymphocytes ratio were the only parameters associated with short PFS in univariate analysis of the parameters studied. However, reduced PFS was independently associated with the proportion of CD20+ B lymphocytes < 0.135% (hazard ratio [HR] = 6.09, *p* = 0.03) and the CD8-to-CD20 lymphocytes ratio ≥ 3.366 (hazard ratio [HR] = 17.82, *p* = 0.0001) (Table 3).

## 3. Discussion

The primary hypothesis of the present study posited that the distinct characteristics of the cellular composition within the tumor microenvironment could be linked to the DoCB of immunotargeted therapy. Our collected data validate this hypothesis. Our study revealed that a high proportion of B lymphocytes and a high CD8-to-CD20 lymphocytes ratio in the tumor microenvironment were associated with a positive response to Pembrolizumab and Lenvatinib therapy. However, the CD8+/CD20+ lymphocyte ratio demonstrated the highest predictive value, indicating that a ratio below 3.366 correlated with extended DoCB. In other words, lower CD8+ lymphocyte counts and/or higher CD20+ lymphocyte counts are linked to the effectiveness of immunotargeted therapy in individuals with advanced or metastatic endometrial cancer. Given that CD8+ lymphocyte counts did not show a direct correlation with treatment response, it is likely that the CD20+ lymphocyte count holds greater significance in this context. It is important to note that when calculating the proportion of B lymphocytes, we excluded tertiary lymphoid structures (TLS), which were found in only 6.9% of patients. The existing data on the role of tumor-associated B cells are inconsistent. In mouse cancer models, B cells have been shown to promote anti-tumor inflammation [13], but they can also inhibit anti-tumor T cell therapy responses [14]. In a study conducted by Riaz et al. on patients with melanoma, it was observed that Anti-PD-1 therapy often leads to an increase in B lymphocytes numbers [15]. Recent evidence also suggests an enrichment in activated B lymphocytes phenotypes, and the contribution of B lymphocytes to TLS formation may facilitate the induction of T cell phenotypes necessary for a response to checkpoint inhibitors [16]. The role of B lymphocytes in the endometrium remains incompletely understood, as highlighted in the review by Shen et al. [17]. Nevertheless, it is plausible that B lymphocytes within tertiary lymphoid structures serve as organized antigen-presenting cells for neighboring T cells, given their heightened expression of activation markers such as CD69, HLA-DR, and CD83 compared to peripheral B lymphocytes [18]. Existing literature does not provide evidence supporting the predictive relevance of B lymphocytes in endometrial cancer immunotherapy. However, research efforts have focused on exploring the predictive significance of B lymphocytes in the immunotherapy of various other malignancies. For instance, a study by Wu Z. et al. identified a subset of B lymphocytes associated with favorable responses to immunotherapy across patients with melanoma, glioblastoma, non-small cell lung cancer, head and neck squamous cell carcinoma, and renal cell carcinoma. Subsequent analysis by the researchers revealed a moderate increase in memory B lymphocytes and a significant decrease in naïve B lymphocytes within tumors that responded to therapy [19]. Another study indicated that the quantity of B lymphocytes within the primary tumor and post-immunotherapy increased among responders compared to non-responders (*p* < 0.05) [20]. When discussing the mechanisms through which B lymphocytes contribute to the efficacy of immunotherapy, it is imperative to reference literature suggesting that B lymphocytes can function as antigen-presenting cells [21]. This observation may elucidate the strong correlation between a higher number of B lymphocytes and the positive outcomes of Pembrolizumab, indirectly implying the presence of an immune response within the tumor. Notably, our search did not yield literature discussing the involvement of B lymphocytes in the mechanisms of action of Lenvatinib. However, there is evidence in the literature that Lenvatinib increased T lymphocytes infiltration into tumors by upregulating the expression of CXCL10 and CCL8 [22]. It is known that macrophage production of CXCL10 amplifies the production of IL-6 by B lymphocytes, leading to plasma cell differentiation [23]. Thus, it can be assumed that both Pembrolizumab and Lenvatinib can indirectly influence the state of B lymphocytes.

Another important finding in our study was that all B lymphocytes from patients with a poor response do not express PD1. This means that these cells are not affected by Pembrolizumab if it is administered to the patient. However, this is not an absolute indication that patients will not respond to therapy, as only six of nineteen responders had CD20+PD1+ B lymphocytes. Nevertheless, this finding may indicate the importance of the functional status of B lymphocytes in endometrial cancer patients. Indirectly, this is confirmed by the data of Horeweg et al., who used single-cell RNA sequencing (scRNA-seq) to show that the stroma of endometrial cancer contains three main clusters of B cells: naïve B cells, (pre-)Germinal Centre (GC)-like cells, and plasma cells [24]. However, in this study, only the intrastromal CD20+ lymphocytes density was significantly associated with a lower risk of recurrence. Our study also showed that a decreased proportion of CD20+ B lymphocytes was associated with progression-free survival (PFS) independently of clinical parameters. The predictive parameter for DoCB in immunotargeted therapy, identified through our multivariate analysis, also emerged as a prognostic factor, independent of the primary clinicopathological parameters conventionally linked to prognosis. In 2023, the International Federation of Gynecology and Obstetrics (FIGO), while revising the staging system for endometrial cancer, deliberated on incorporating molecular classification and assessing lymphovascular invasion and the size of lymph node metastases [25]. Analysis by The Cancer Genome Atlas (TCGA) resulted in a comprehensive categorization of endometrial cancer into four distinct genomic subgroups: POLE ultramutated (POLEmut), hypermutated with MSI-H or MMRd, p53 mutant (p53abn), and nonspecific [26]. Each subgroup corresponds to a varied prognosis, with POLEmut cases typically exhibiting the most favorable outlook and p53abn cases the least, while the remaining two categories fall in between. Although our study did not delineate the molecular subtypes of endometrial cancer, the MMS/pMMR status excluded our patients from the hypermutated subtype. Literature findings underscore the prognostic importance of tumor microenvironment infiltration severity across different molecular subtypes. For instance, elevated levels of CD3+CD8+ and CD3+CD8- tumor-infiltrating lymphocytes in POLE tumors, along with increased counts of B cells and plasma cells in the stromal regions of POLE tumors, have been noted. Recent research also highlights significantly higher B lymphocyte counts in low-risk groups compared to high-risk groups in endometrial cancer [27]. These insights further validate the prognostic significance of the CD8-to-CD20 lymphocyte ratio identified in our study concerning advanced or metastatic endometrial cancer.

Both parameters we identified can be readily utilized in routine clinical practice by pathologists. While our study employed immunofluorescence, the assessment of cell fractions can also be carried out using the standard immunohistochemistry. However, it is important to note that without CD20+ B lymphocytes present in the tumor, calculating the CD8-to-CD20 lymphocytes ratio would not be feasible. Nevertheless, if the proportion of CD20+ B lymphocytes is zero, it can still be utilized for predictive and prognostic assessments. Thus, the proportion of CD20+ B lymphocytes and the CD8-to-CD20 lymphocytes ratio in the stroma of endometrial cancer serves as both a predictive marker of response to immunotargeted therapy and a prognostic factor for progression-free survival in patients.

## 4. Materials and Methods

### 4.1. Patients

This study enrolled 28 patients diagnosed with recurrent or metastatic endometrial cancer and treated them with a combination of anti-PD-1 therapy and multi-targeted tyrosine kinase inhibitor therapy. The MSS and/or pMMR status of all patients was determined through genetic testing or immunohistochemical analysis. The treatment was administered as a 1–6 line therapy in the Department of Gynecology, Cancer Research Institute, Tomsk National Research Medical Center, Irkutsk Regional Oncology Center, Novosibirsk Regional Clinical Oncology Center, Altai Regional Oncological Dispensary, Krasnoyarsk Regional Oncological Center, and Yakut Republican Oncology Center. Follow-up was conducted for a minimum of 7 months. Patients received standard doses of Pembrolizumab plus Lenvatinib according to the recommended schedules until disease progression or unacceptable toxicity occurred. Toxicity was monitored and recorded using the Common Terminology Criteria for Adverse Events (version 4.0), assessed on day 1 of each treatment cycle until the end of the treatment period.

The inclusion criteria for the study were the following: age greater than 18 years, a histologically confirmed diagnosis of endometrial cancer, and an ECOG status ranging from 0 to 2 points. Exclusion criteria included autoimmune diseases, systemic immunosuppression, and significant comorbidities. The study assessed progression-free survival (PFS), overall survival (OS), and duration of clinical benefit (DoCB). PFS was defined as the time from the start of immunotherapy to the first documented tumor progression or death from any cause. OS was defined as the interval between the start of immunotherapy and death from any cause.

Response evaluation was performed monthly until disease progression using the immune-related response evaluation criteria in solid tumors (iRECIST). Responses were categorized as complete or partial response, stable disease, or progressive disease. DoCB was used to classify patients as responders (those with complete or partial response and stable disease) or non-responders (progressors) after 6 months of therapy. The DoCB is defined as the time from start treatment to disease progression or death in patients who achieve complete response, partial response, or stable disease for 24 weeks. The study was conducted in accordance with the Declaration of Helsinki and adhered to good clinical practice guidelines. All patients provided signed informed consent (protocol №18, 25 August 2023).

### 4.2. TSA-Associated Multiplex Immunofluorescence

The study utilized TSA-associated multiplex immunofluorescence to analyze the composition of immune cells in the microenvironment and their expression of PD-1. The Vectra^®^ 3.0 automated quantitative imaging system from Akoya Biosciences, Marlborough, MA, USA, was employed for image acquisition, and the inForm^®^ 3.3 software, also from Akoya Biosciences, USA, was used for image analysis. The InForm^®^ phenotyping software employed a machine learning approach for cell identification and classification. For each sample, the entire preparation was analyzed, excluding areas with artifactual staining or poor quality. The following panel of antibodies was used: anti-CD8 (clone SP57, Ventana, Oro Valley, AZ, USA), anti-PD-1 (clone NAT105, Cell Marque, Rocklin, CA, USA), anti-CD20 (clone L26, Leica, Deer Park, IL, USA), anti-CD163 (clone 10D6, Diagnostic BioSystems, Pleasanton, CA, USA), and anti-FoxP3 (clone 236A/E7, Invitrogen, Waltham, MA, USA). The nuclei were stained with DAPI and mounted in ProLong Gold Antifade Mountant (Thermo Fisher, Waltham, MA, USA). The proportion of cells (CD8+ cytotoxic lymphocytes, CD20+ B lymphocytes, FoxP3+ T-regulatory (Treg) lymphocytes, CD163+ macrophages) was calculated in 10 fields (magnification 200×) as the percentage of all cells in the tumor stroma (Figure 6). All stroma within the tumor node were included in the analysis. The surrounding stromal tissue that did not contain tumor cells was deemed unrelated to the tumor and was excluded in the analysis as well as tertiary lymphoid structures.

### 4.3. Statistical Analysis

Statistical analyses were conducted using Prism 10 from GraphPad. The Mann–Whitney test was employed for comparing independent nonparametric variables, while the Fisher exact test was used for categorical variables. To evaluate the predictive performance of the trait, ROC analysis was performed. Accuracy of the criterion was determined by calculating the Area Under the Curve (AUC), confidence interval (CI), sensitivity, and specificity values. Univariate and multivariate Cox regression analysis was performed to for the hazard ratio assessment. The Kaplan–Meier curve was used for survival analysis (log-rank criterion). The Bootstrap technique (*n* = 1000) was then applied to test the efficiency of the estimated non-linear simple regression models. All *p* values were two-sided, and a significance level of <0.05 was considered statistically significant.

## 5. Conclusions

Our study demonstrates that the proportion of CD20+ B lymphocytes and the CD8-to-CD20 lymphocytes ratio in the stroma of endometrial cancer serves as a predictive marker for response to immunotargeted therapy as well as a prognostic factor for progression-free survival in patients. Although the sensitivity and specificity of this marker do not reach 100%, its utilization can still improve the selection of candidates for immunotargeted therapy.

## Figures and Tables

**Figure 1 ijms-25-03933-f001:**
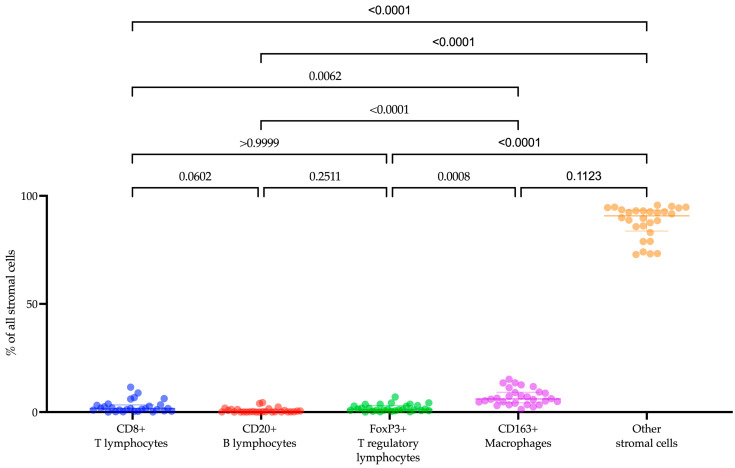
The proportion of CD8+ T lymphocytes, CD20+ B lymphocytes, FoxP3+ T lymphocytes, CD163+ macrophages, and other stromal cells within the tumor microenvironment in endometrial cancer patients.

**Figure 2 ijms-25-03933-f002:**
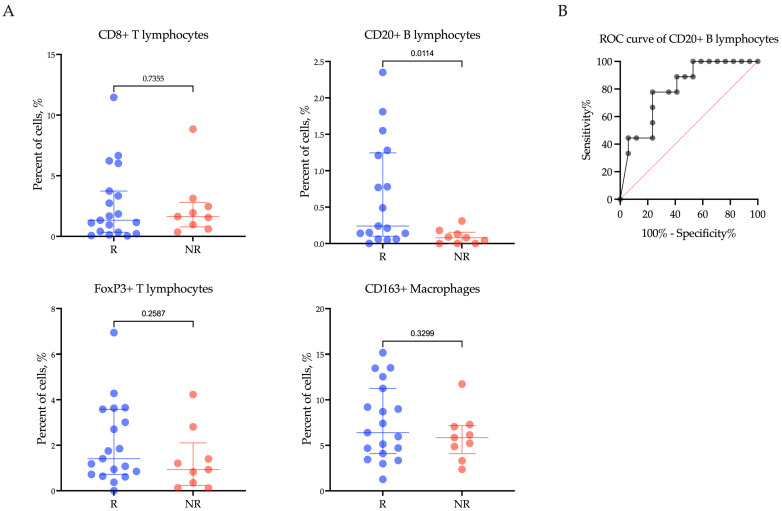
The proportion of CD8+ T lymphocytes, CD20+ B lymphocytes, FoxP3+ T lymphocytes, and CD163+ macrophages (**A**) within the tumor microenvironment in endometrial cancer patients according to response to treatment with Pembrolizumab plus Lenvatinib, and (**B**) ROC analyses of the predictive value of CD20+ B lymphocytes. Rs: responders; NRs: non-responders.

**Figure 3 ijms-25-03933-f003:**
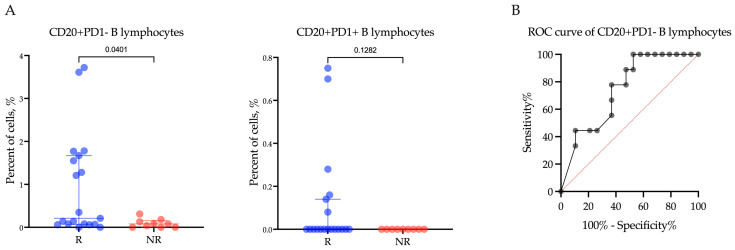
The proportion of CD20+PD1− B lymphocytes and CD20+PD1+ B lymphocytes (**A**) within the tumor microenvironment in endometrial cancer patients according to response to treatment with Pembrolizumab plus Lenvatinib, and (**B**) ROC analyses of the predictive value of CD20+PD1− B lymphocytes. Rs: responders; NRs: non-responders.

**Figure 4 ijms-25-03933-f004:**
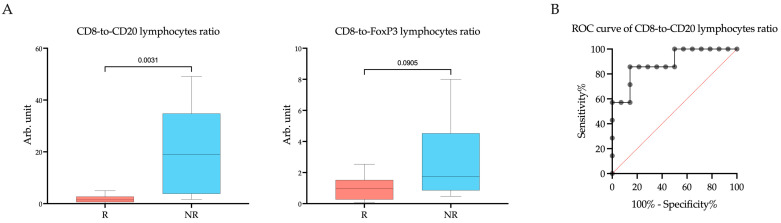
The CD8-to-CD20 lymphocytes ratio and CD8-to-FoxP3 lymphocytes ratio (**A**) within the tumor microenvironment in endometrial cancer patients according to response to treatment with Pembrolizumab plus Lenvatinib, and (**B**) ROC analyses of the predictive value of the CD8-to-CD20 lymphocytes ratio. Rs: responders; NRs: non-responders.

**Figure 5 ijms-25-03933-f005:**
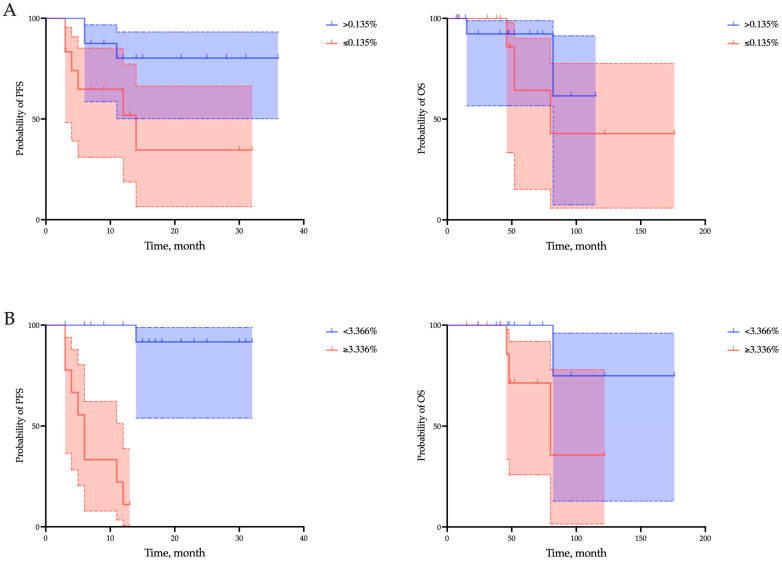
Survival analyses of progression-free survival (PFS) and overall survival (OS) regarding the proportion of CD20+ B lymphocytes (**A**) and the CD8-to-CD20 lymphocytes ratio (**B**) within the tumor microenvironment in endometrial cancer patients. Overall survival is defined as the interval between the start of immunotherapy and death from any cause. Progression-free survival is defined as the time from the start of immunotherapy to the first documented tumor progression or death from any cause.

**Figure 6 ijms-25-03933-f006:**
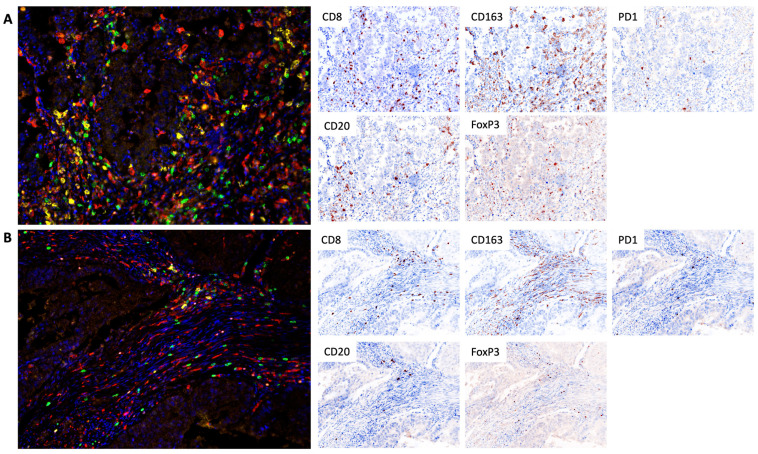
Tumor infiltration of CD8+ cytotoxic lymphocytes, CD20+ B lymphocytes, FoxP3+ T-regulatory (Treg) lymphocytes, and CD163+ macrophages in the cohort of responders (**A**) and non-responders (**B**). In the multiplex immunofluorescence images CD8+ cytotoxic lymphocytes have a green pseudo-color, CD20+ B lymphocytes are yellow, CD163+ macrophages are red, Treg lymphocytes are orange, PD1 expression is white, and DAPI (nuclear counterstain) is blue. Separate markers with nuclear are presented as IHC simulations, which can be performed by Inform software based on obtained multiplex staining data. TSA-associated multiplex immunofluorescence, 200×.

**Table 1 ijms-25-03933-t001:** Clinical and pathological parameters of patients with recurrent or metastatic endometrial cancer.

Parameter, Abs. (%)		
Age (years)		
FIGO	IA	3/28 (10.7)
	IB	5/28 (17.8)
	II	7/28 (25.0)
	IIIA	5/28 (17.8)
	IIIC	3/28(10.7)
	IVB	5/28 (17.8)
Histological type	Endometrioid	21/28 (75.0)
	Papillary serous	7/28 (25.0)
Grade	I	6/28 (21.4)
	II	11/28 (39.3)
	III	8/28 (28.6)
	No data	3/28 (10.7)
Myometrium involvement	None	1/28 (3.6)
	<50%	11/28 (39.3)
	>50%	16/28 (57.1)
Lymphovascular invasion (LVI)	No	16/28 (57.1)
	Yes	12/28 (42.8)
Metastasis	No	7/28 (25.0)
	Yes	21/28 (75.0)

**Table 2 ijms-25-03933-t002:** Univariate non-linear simple regression analysis and Bootstrap validation of the prognostic significance of clinicopathological and tumor microenvironment parameters in patients with endometrial cancer.

Variables	Univariate Analysis		Bootstrap
	OR (95%CI)	*p* Value	Bias
FIGO (I–II vs. III–IV)	1.11 (0.57 to 3.67)	0.29	0.02
Histological type (Endometrioid vs. Papillary serous)	1.66 (0.50 to 8.32)	0.44	0.05
Metastasis (Yes vs. No)	0.96 (0.46 to 1.63)	0.59	0.34
Grade (1–2 vs. 3)	1.97 (0.84 to 6.76)	0.36	0.09
Myometrial invasion (Yes vs. No)	1.09 (0.05 to 1.38)	0.93	0.42
Lymphovascular invasion (Yes vs. No)	1.55 (0.29 to 2.19)	0.74	0.55
Proportion of CD20+ (>0.135 vs. ≤0.135)	2.56 (1.12 to 6.80)	0.004	0.001
CD8-to-CD20 lymphocytes ratio (<3.366 vs. ≥3.366)	8.21 (2.76 to 23.10)	0.0001	0.001

OR: odds ratio, 95%CI: 95% confidence interval.

**Table 3 ijms-25-03933-t003:** Univariate and multivariate Cox regression analysis of the prognostic significance of clinicopathological and tumor microenvironment parameters in patients with endometrial cancer.

Variables	Univariate Analysis		Multivariate Analysis	
	HR (95%CI)	*p* Value	HR (95%CI)	*p* Value
FIGO (I–II vs. III–IV)	1.32 (0.64 to 15.23)	0.45	1.27 (0.36 to 10.90)	0.56
Histological type (Endometrioid vs. Papillary serous)	0.79 (0.45 to 1.73)	0.78	1.08 (0.12 to 3.97)	0.53
Metastasis (Yes vs. No)	1.45 (0.34 to 9.83)	0.64	1.62 (0.33 to 17.25)	0.57
Grade (1–2 vs. 3)	3.26 (0.84 to 13.41)	0.08	4.94 (0.79 to 11.42)	0.14
Myometrial invasion (Yes vs. No)	0.78 (0.21 to 3.18)	0.72	0.47 (0.15 to 3.28)	0.75
Lymphovascular invasion (Yes vs. No)	1.27 (0.31 to 4.84)	0.71	0.61 (0.10 to 2.84)	0.53
Proportion of CD20+ (>0.135 vs. ≤0.135)	4.01 (1.10 to 19.1)	0.04	6.09 (1.31 to 63.12)	0.03
CD8-to-CD20 lymphocytes ratio (<3.366 vs. ≥3.366)	26.84 (5.67 to 126.9)	0.0001	17.82 (3.93 to 117.11)	0.0001

## Data Availability

The data presented in this study are available on request from the corresponding authors.

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
