# Peer review of "Predicting Response to Immunotargeted Therapy in Endometrial Cancer via Tumor Immune Microenvironment: A Multicenter, Observational Study"

_ijms, 2024, doi:10.3390/ijms25073933_

Round 1
Reviewer 1 Report
Comments and Suggestions for Authors
Thank you for your interesting paper. If applicable to clinical settings, the findings could be of very high importance.
1. Parameters in table 1 are well listed. Were all these parameters included in the Univariate Analysis / Multivariate Analysis of table 2?
2. The data seem to suggest that higher B cell was associated with better response. However, line 99-100 said '2.2. Low Levels of CD20+ B Cells Are Associated with Response to Pembrolizumab plus Lenvatinib Treatment'. Please clarify the meaning of this title
3. The cohort is small, but in view of difficult patient recruitment, it is acceptable.
4. How was the cutoff 0.135% determined (line 117)?
5. Was all stroma area included for analysis ? If only selected, how to select ? How was stroma defined and determined? (line 249) The image analysis sectino needs more detail to make method reproducible.
Comments on the Quality of English LanguageThere is no major issue with the language.
Lines 79-81 seem unnecessary.
Author Response
We appreciate the reviewer's thorough analysis of our manuscript and the valuable feedback provided. We have made efforts to clarify the paper and trust that its quality has been enhanced as a result.
Q: 1. Parameters in table 1 are well listed. Were all these parameters included in the Univariate Analysis / Multivariate Analysis of table 2?
A: Additional FIGO parameters (I-II vs. III-IV) and histological types (Endometrioid vs. Papillary serous) were included in the Univariate Analysis and Multivariate Analysis. The revised data are detailed in Table 2.
Q: 2. The data seem to suggest that higher B cell was associated with better response. However, line 99-100 said '2.2. Low Levels of CD20+ B Cells Are Associated with Response to Pembrolizumab plus Lenvatinib Treatment'. Please clarify the meaning of this title
A: Line 100 is inaccurate
Q: 3. The cohort is small, but in view of difficult patient recruitment, it is acceptable.
A: We appreciate your emphasis on this matter. Indeed, the patient cohort under study is currently quite restricted, and the insights from our experience with advanced endometrial cancer in Siberia and the Russian Far East, as outlined in this paper, have been gathered over a two-year timeframe. Despite the limited sample size, we employed pertinent statistical techniques, and the results presented represent the second iteration of the analysis. The initial interim analysis was conducted on a truncated sample, and the outcomes were replicated with complete precision. These factors collectively provide a basis for considering the results reliable.
Q: 4. How was the cutoff 0.135% determined (line 117)?
A: ROC analysis was performed to determine a cut off responders from non-responders with maximum sensitivity and specificity values
Q: 5. Was all stroma area included for analysis ? If only selected, how to select ? How was stroma defined and determined? (line 249) The image analysis section needs more detail to make method reproducible.
A: All stroma within the tumor node, except for extra-tumor stroma and tertiary lymphoid structures, were included in the analysis. The description has been expanded and added to the corresponding section.
Q: Lines 79-81 seem unnecessary.
A: Lines 79-81 was deleted, this is a technical inaccuracy, template lines have been left out, please apologize
Reviewer 2 Report
Comments and Suggestions for Authors
It would be important to have a stratification during the treatment which can be readily assessed both in scientific and cost-effective manner. The authors thought process to understand the responders and non-responders through immune response in TME is a good strategy.
One of the biggest caveats of the study is having a limited sample size, also the immune cell quantification was performed only in the TME region. But it would have been more informative if the information is also collected using the blood samples.
Only a very limited number (n=7) of respondents (n=19) have shown the CD20+ B lymphocyte upregulation compared to non-respondents (n=9). The trend might be significant because of few samples but more sample number is needed to draw the conclusion.
CD163+ macrophages might be significantly higher in respondents over NR’s, a single data point has changed the trend. Data is trying to show that, combination of events might be happening to change the outcomes.
Please share your insights about the reasons for increase in CD20+ B lymphocytes being not translated to change in T-lymphocytes but changed the overall outcome.
In Fig2A some patients from respondents have higher % of CD20+ and PD1- B-lymphocytes (Fig-2A), Is it the same patient samples that have shown the higher expression of CD20+ B lymphocytes in Fig1B. Could you try connecting the pembrolizumab effect on higher CD20+ B-lymphocyte population based on this observation?
CD20+ B-lymphocyte cut off at 0.135% is well translated in PFS analysis but not in OS, can you please suggest what might have affected the outcome. Does having a different cut-off might help to understand more towards the OS outcome which could impact in taking the decision?
Please cite this manuscript “Lenvatinib plus Pembrolizumab for Advanced Endometrial Cancer”
Comments on the Quality of English LanguageOverall quality of the manuscript can be improved, especially the discussion should be written in a more elaborative manner. Results need to be explained more clearly.
Author Response
We appreciate the reviewer's thorough analysis of our manuscript and the valuable feedback provided. We have made efforts to clarify the paper and trust that its quality has been enhanced as a result.
It would be important to have a stratification during the treatment which can be readily assessed both in scientific and cost-effective manner. The authors thought process to understand the responders and non-responders through immune response in TME is a good strategy.
Q: One of the biggest caveats of the study is having a limited sample size, also the immune cell quantification was performed only in the TME region. But it would have been more informative if the information is also collected using the blood samples.
A: We appreciate your focus on this aspect. It is true that the patient cohort under study is currently quite limited, and the expertise with advanced endometrial cancer in Siberia and the Russian Far East, as presented in this paper, has been accumulated over a two-year period. Despite the constrained sample size, we applied pertinent statistical methodologies, and the results presented constitute the second iteration of the analysis. The initial interim analysis was conducted on a truncated sample, and the outcomes were replicated with 100% accuracy. These factors collectively support their reliability.
The tumor microenvironment was selected as the focal point of our study because it is believed to exert the most significant influence on tumor cells, determining their malignant potential among other factors, rather than blood immune factors. Moreover, immunotargeted therapies predominantly aim at the tumor microenvironment. Nonetheless, we acknowledge the importance of assessing systemic immunity parameters as well.
Q: Only a very limited number (n=7) of respondents (n=19) have shown the CD20+ B lymphocyte upregulation compared to non-respondents (n=9). The trend might be significant because of few samples but more sample number is needed to draw the conclusion.
A: We included the previously missing data on CD20+ B lymphocyte levels for two patients and supplemented the results with bootstrap mathematical validation data. The analysis revealed significant prognostic value when the sample size was expanded to 1000 cases.
Q: CD163+ macrophages might be significantly higher in respondents over NR’s, a single data point has changed the trend. Data is trying to show that, combination of events might be happening to change the outcomes.
A: As part of our statistical analysis process, we examine the sample for outliers. For the proportion of CD163+ macrophages, no outliers were identified initially. However, we validated this by excluding the data point with a value of 11.74% from the sample, resulting in a confidence level of differences at 0.2828.
Q: Please share your insights about the reasons for increase in CD20+ B lymphocytes being not translated to change in T-lymphocytes but changed the overall outcome.
A: The proportions of T-lymphocytes and B-lymphocytes are not biologically linked. They have distinct origins within the tumor, leading to expected variations in their clinical significance. Your inquiry inspired us to consider not only the individual population counts but also their ratio. This approach enabled us to identify a more precise biomarker for assessing the response to immunotargeted therapy in endometrial cancer patients.
Q: In Fig2A some patients from respondents have higher % of CD20+ and PD1- B-lymphocytes (Fig-2A), Is it the same patient samples that have shown the higher expression of CD20+ B lymphocytes in Fig1B. Could you try connecting the pembrolizumab effect on higher CD20+ B-lymphocyte population based on this observation?
A: Thank you for your suggestion. We have reevaluated the cell counts by expanding the number of fields of view as per your suggestion, and we are now presenting the updated data. The discussion section has been significantly expanded to incorporate the point you raised.
Q: CD20+ B-lymphocyte cut off at 0.135% is well translated in PFS analysis but not in OS, can you please suggest what might have affected the outcome. Does having a different cut-off might help to understand more towards the OS outcome which could impact in taking the decision?
A: We conducted further analyses to assess the prognostic significance of all the parameters studied in relation to overall survival. However, none of these analyses provided us with adequate sensitivity and specificity to predict the duration of overall survival for patients. This limitation may be closely tied to the varying time intervals until the initial progression, at which point immunotargeted therapy is initiated. This period can range from several years in some patients to just a few months in others.
Q: Please cite this manuscript “Lenvatinib plus Pembrolizumab for Advanced Endometrial Cancer”
A: The above reference has been added to the text of the manuscript and the discussion section of the results has been added.
Q: Overall quality of the manuscript can be improved, especially the discussion should be written in a more elaborative manner. Results need to be explained more clearly.
A: We have significantly enriched the discussion section by including our insights into the mechanisms through which the studied cells are implicated in the response to pembrolizumab therapy.
Reviewer 3 Report
Comments and Suggestions for Authors
It is well known that MSI status was approved as a gold standard for immunotherapy application independently on the histology and biology of the tumor being one of the agnostic diagnostic instruments. Although surrogate markers (e.g. immunohistochemistry) do not work equivalently. However, the endometrium can be tested through MMR IHC marker effectively not all dMMR patients can be a beneficial candidate for immunotherapy and addition methods for patients’ stratification are required. The authors analyze different immune cells to evaluate the best prognostic marker and demonstrate the results with different combination of the tested panels. The multiplex technology application facilitates the demonstration of different markers expression simultaneously. I suppose that more multiplex images should be provided to demonstrate the comprehensive analysis. In addition, it would be better to demonstrate or at least to describe how evaluated markers could be implemented in routine practice of pathology department.
Author Response
We appreciate the reviewer's thorough analysis of our manuscript and the valuable feedback provided. We have made efforts to clarify the paper and trust that its quality has been enhanced as a result.
It is well known that MSI status was approved as a gold standard for immunotherapy application independently on the histology and biology of the tumor being one of the agnostic diagnostic instruments. Although surrogate markers (e.g. immunohistochemistry) do not work equivalently. However, the endometrium can be tested through MMR IHC marker effectively not all dMMR patients can be a beneficial candidate for immunotherapy and addition methods for patients’ stratification are required. The authors analyze different immune cells to evaluate the best prognostic marker and demonstrate the results with different combination of the tested panels.
Q: The multiplex technology application facilitates the demonstration of different markers expression simultaneously. I suppose that more multiplex images should be provided to demonstrate the comprehensive analysis.
A: We have added additional multiplex analysis images to the manuscript that more strongly illustrate the differences we found in responders and non-responders.
Q: In addition, it would be better to demonstrate or at least to describe how evaluated markers could be implemented in routine practice of pathology department.
A: We have covered this aspect in the discussion section, expanding on it considerably.
Reviewer 4 Report
Comments and Suggestions for Authors
The paper “Predicting Response to Immunotargeted Therapy in Endometrial Cancer via Tumor Immune Microenvironment: a Multicenter, Observational Study” analyzed the proportion of CD8+ T lymphocytes, CD20+ B lymphocytes, FoxP3+ T regulatory lymphocytes, and CD163+ macrophages in tumor samples prior to immunotargeted therapy. Here are still some shortcomings that need to be further improved or explained.
Comments:
Q1. Line 97-98, the tittle of Table 1 was missing.
Q2. What are the main compositions of tumor-infiltrating immune cells? The average contents of the several shown in the text were all less than 5%. Were the remaining 90% all tumor cells?
Q3. The percentages of CD20+ B lymphocytes were too low, are these data statistically significant?
Q4. Why did not choose CD20+PD1-B lymphocytes as the predictive and prognostic factor?
Q5. Are there necessary links between responders (non-responders) and progression-free survival (overall survival)?
Q6. The paper lacks sufficient valid data and the evidence provided is inadequate. It is difficult to accurately determine such low B cell proportions.
Author Response
We appreciate the reviewer's thorough analysis of our manuscript and the valuable feedback provided. We have made efforts to clarify the paper and trust that its quality has been enhanced as a result.
The paper “Predicting Response to Immunotargeted Therapy in Endometrial Cancer via Tumor Immune Microenvironment: a Multicenter, Observational Study” analyzed the proportion of CD8+ T lymphocytes, CD20+ B lymphocytes, FoxP3+ T regulatory lymphocytes, and CD163+ macrophages in tumor samples prior to immunotargeted therapy. Here are still some shortcomings that need to be further improved or explained.
Comments:
Q1. Line 97-98, the tittle of Table 1 was missing.
A: Lines 97-98 was deleted; title of Table 1 was added. This is a technical inaccuracy, template lines have been left out
Q2. What are the main compositions of tumor-infiltrating immune cells? The average contents of the several shown in the text were all less than 5%. Were the remaining 90% all tumor cells?
A: We have added data on compositions of tumour microenvironment to address this question.
Q3. The percentages of CD20+ B lymphocytes were too low, are these data statistically significant?
A: The magnitude of numbers has never been an obstacle to finding a statistical difference between two sets of data. We use the nonparametric Mann-Whitney criterion for analysis, which is a ranking criterion, i.e. it does not depend on the magnitude of the number, because observations from both groups are combined and ranked, with matching values being assigned a mean rank.
Q4. Why did not choose CD20+PD1-B lymphocytes as the predictive and prognostic factor?
A: Based on the best sensitivity and specificity we selected the total proportion of CD20+ B lymphocytes as a predictive marker. In the course of additional analysis we found a more significant parameter for response prediction. These data have been added to the text of the article.
Q5. Are there necessary links between responders (non-responders) and progression-free survival (overall survival)?
A: when we calculated progression-free survival or overall survival, the stratification of patients was done not by responder and non-responder, but by cell count or its ratio.
Q6. The paper lacks sufficient valid data and the evidence provided is inadequate. It is difficult to accurately determine such low B cell proportions.
A: We performed additional analyses, including statistical analyses, to improve the validity of the data presented. We added additional data to the manuscript.
Round 2
Reviewer 2 Report
Comments and Suggestions for Authors
Authors have addressed most of the comments.
Author Response
We express our deep appreciation to the reviewer for analyzing our manuscript. We would like to emphasize once again that despite the small number of patients included in the study, we believe that our data are reliable. This is evidenced by the continued significance of differences when varying the number of patients in the sample, e.g., by different clinical features. In addition, we performed statistical analyses that allow us to work with small samples, and appropriate criteria were used. Additionally, a logistic regression analysis was performed, followed by a variation that showed minimal bias. All these factors together, in our opinion, confirm the reliability of our conclusions.
Reviewer 4 Report
Comments and Suggestions for Authors
Major concerns were not well addressed.
Author Response
We are very grateful for the analysis of our manuscript and provide extended responses to the reviewer's comments:
Q2. What are the main compositions of tumor-infiltrating immune cells? The average contents of the several shown in the text were all less than 5%. Were the remaining 90% all tumor cells?
A: We have added data on compositions of tumour microenvironment to address this question.
Extended Answer: The average number of those immune cells that we phenotyped in the tumour microenvironment was about 10%. Most of the remaining cells were fibroblasts, which is very clear from Figure 6B, as well as macrophages (since the CD163 marker detects predominantly M2 macrophages), other lymphocytes, dendritic cells, endotheliocytes, and others.
Q5. Are there necessary links between responders (non-responders) and progression-free survival (overall survival)?
A: when we calculated progression-free survival or overall survival, the stratification of patients was done not by responder and non-responder, but by cell count or its ratio.
Extended Answer: Since we stratified patients by Duration of clinical benefit (DoCB), which is defined as the time from start of treatment to disease progression or death in patients who achieve complete response, partial response, or stable disease for 24 weeks, if we were to estimate progression-free survival in responders and non-responders, no cases of disease progression would be found in responders and we could then speak of a linear relationship between the parameters. However, by estimating progression-free survival we stratified patients by the threshold values found, thus assessing how the proportion of cells that can be detected much more earlier than the patient starts receiving treatment can be an indicator of the duration of progression-free survival.
Q6. The paper lacks sufficient valid data and the evidence provided is inadequate. It is difficult to accurately determine such low B cell proportions.
A: We performed additional analyses, including statistical analyses, to improve the validity of the data presented. We added additional data to the manuscript.
Extended Answer: We would like to emphasize once again that despite the small number of patients included in the study, we believe that our data are reliable. This is evidenced by the continued significance of differences when varying the number of patients in the sample, e.g., by different clinical features. In addition, we performed statistical analyses that allow us to work with small samples, and appropriate criteria were used. Additionally, a logistic regression analysis was performed, followed by a variation that showed minimal bias. All these factors together, in our opinion, confirm the reliability of our conclusions.
Round 3
Reviewer 4 Report
Comments and Suggestions for Authors
No additional questions.